# Agent-based modeling of macrophage-fibroblast interactions in the immune response to biomaterials

Jennifer Riccio[1☉], Luca Presotto[1☉*], Shir Bahiri[2], Liad Doniza[2], Donato Inverso[3], Laura Sironi[1], Uri Nevo[2], Giuseppe Chirico[1]

**1** Department of Physics, Università degli Studi di Milano-Bicocca, Milan, Italy, **2** Department of Biomedical Engineering, Tel Aviv University, Tel Aviv, Israel, **3** Division of Immunology, Transplantation and Infectious Diseases, Vita-Salute San Raffaele University, Milan, Italy

☉ These authors contributed equally to this work.

* luca.presotto@unimib.it

**Data availability statement:** All data and code used for running experiments, model fitting, and

## Abstract

Foreign body reaction (FBR) denotes the reaction to the implantation of a biomaterial into the body. It triggers cascades of responses in the tissue and involves different cell types including, among others, macrophages, fibroblasts and endothelial cells. Macrophages regulate the inflammatory and healing processes. They exhibit a variety of functional phenotypes (or states) induced by the stimulus they receive and the microenvironment. This polarization process is governed by chemical mediators, known as cytokines, that are secreted by the macrophage itself and induce cellular activation and recruitment. Cytokines determine the macrophage phenotype within a heterogeneous range that spans between two extremes: pro-inflammatory or $M_1$ and anti-inflammatory (or pro-healing) or $M_2$.

Fibroblasts are recruited in response to cytokine secretion and play a crucial role in tissue remodeling. These cells generate key components of the extracellular matrix (ECM), such as elastin, fibrin, and collagen, and have the ability to isolate the implanted biomaterial from surrounding tissue by encapsulating it within a fibrotic layer. The formation of this fibrotic capsule is a major factor contributing to the failure of many biomaterials.

Macrophage and fibroblasts interact in tissues both in physiological and pathological conditions. One of the major signaling factor is the colony-stimulating factor 1 (CSF1) and its specific receptor (CSF1R). In the past, simulation works have focused only on the description of the phenotype transition from $M_1$ to $M_2$ for macrophages, possibly connected to physiological and pathological conditions (e.g. hypoxia), but neglecting the relevant macrophage/fibroblast interaction.

Our long term aim is to exploit an agent-based (AB) modeling approach to develop a predictive digital twin for simulating the response over time of the cell populations involved in a FBR. Our first step in this direction is the explicit introduction of the interaction between macrophages and fibroblasts.

To achieve this goal, we consider here at first the existing ordinary differential equation (ODE) and AB models, that simulate intra- and inter-cellular dynamics for macrophages,

plotting is available on a GitHub repository at
https://github.com/lucaxx85/Fibro_ABM.

**Funding:** This work was supported by the
European Union, FET-OPEN project IN2SIGHT,
Grant Agreement n° 964481, awarded to DI, UN
and GC, and by the European Union -
NextGenerationEU through the Italian Ministry
of University and Research under PNRR -
M4C2-I1.3 Project PR_00000019 "HEAL
ITALIA", CUP J33C22002920006, to GC. The
funders had no role in study design, data
collection and analysis, decision to publish, or
preparation of the manuscript. The views and
opinions expressed are those of the authors
only and do not necessarily reflect those of the
European Union or the European Commission.
Neither the European Union nor the European
Comission can be held responsible for them.

**Competing interests:** The authors have
declared that no competing interests exist.

respectively. We validate them against *in vitro* data taken from experiments that recapitulate the reaction to a pathogen and *in vivo* data taken from the literature. This approach highlights a better agreement of the AB model over the ODE models taken into account in our study. Therefore, we propose a more advanced and comprehensive simulation platform based on AB modeling, which also includes fibroblasts and their mutual interaction with macrophages, as well as fibrosis resulting from the implantation of a biomaterial, allowing us to simulate *in vivo* scenarios. We validate this tool on experimental results from the literature finding a remarkable agreement. The application of this extended AB model allows us to replicate the kinetics of the cell populations involved, including, among others, the effect of different types of stimulus, chemotaxis, recruitment, and formation of the fibrotic capsule typical of the chronic FBR.

## 1 Introduction

Any biomaterial implanted into the body is recognized by the immune system as a foreign body which triggers a cascade of reactions that involve different cell populations [1]. Within seconds of implantation, proteins are adsorbed to the surface of the implant, thus becoming a provisional matrix. Within minutes of implantation, neutrophils migrate into the area and begin to release factors which promote the progression of the inflammatory process. Within a few hours of implantation, the neutrophils give way to a population of macrophages, attracted to the biomaterial surface or resident in the tissue. They represent the core of the inflammatory response and show different states of activation (or phenotypes) depending on the local conditions [2]. This polarization process steers macrophages towards pro-inflammatory ($M_1$-like) or anti-inflammatory or pro-healing ($M_2$-like) phenotypes [3], thus allowing the transition from an acute inflammation state to a chronic resolution and/or to tissue regeneration. It must be noted that the possibility for macrophages to express a continuous palette of phenotypes, ranging from $M_1$ to $M_2$, is now widely recognized [4–6] and supported by models [4]. This situation can be modeled at first approximation by assuming the existence of an intermediate state exhibiting both $M_1$- and $M_2$-like characteristics.

The secretion of anti-inflammatory cytokines and growth factors by macrophages attracts fibrocytes to the surface of the biomaterial and induces their activation. These activated fibroblasts adhere to the surface of the implant and begin depositing layers of extracellular matrix (ECM) proteins. The movement of macrophages is governed by a chemotactic migration guided by inflammation from surrounding tissue to the region of implant [7]. Over a period of weeks to months, the combination of macrophages, fibroblasts, and ECM creates a new tissue, the fibrotic capsule, which envelops the implant and introduces various mechanical cues influencing cellular functions.

Temporal models such as ordinary differential equation (ODE) and agent-based (AB) approaches, are well suited for capturing the dynamic plasticity of macrophages and fibroblasts. In this work, we started by considering the ODE models developed in [8] and in [3], with the latter representing an extended version of the former, to simulate single-cell kinematics. Each model includes a set of differential equations, describing variations in the concentration of biochemical species involved in the described subcellular pathways. The parameters used here are adapted from literature, assumed or estimated from experimental results.

Minucci et al. [3] also proposed an AB scheme, where individual cells are simulated as entities endowed with a state variable (referred to as activation variable) that defines the cell

phenotype and determines its fate. Every macrophage may diffuse in a bidimensional grid according to simplified rules, modeled with fewer equations and parameters than the ODE counterpart. This approach captures the spatiotemporal dynamics at a collective cellular scale and is able to take into account the heterogeneity of the cellular response.

We first present a straightforward comparison of the ODE and AB models from the literature [3,8] with experimental data on the polarization of macrophages treated with pro-inflammatory cytokines *in vitro*. This analysis suggests that the AB scheme more closely reproduces the data than the ODE approaches. The substantial innovations of our contribution are the inclusion of a second species of cells, namely fibrocytes and fibroblasts, the simulation of the interaction of these with macrophages and the inclusion of the chemotactic movement of macrophages and fibrocytes in the original AB model proposed by Minucci et al. [3]. We also took into consideration realistic lifespans for these cells and incorporated the possibility of recruitment of new cells (macrophages, fibrocytes and fibroblasts), depending on the level of secreted cytokines. This step is essential to model the processes that lead to the fibrotic reaction and allows us to include the interaction of the simulation volume with the rest of the organism.

In addition to the description and internal validation of the algorithm, we provide a direct comparison with experimental data, focusing on the secretion of pro- and anti-inflammatory cytokines and on the distribution of cells around the biomaterial. The resulting AB model was then tested by simulating an *in vivo* scenario, where cells were treated with a pathological stimulus, like lipopolysaccharide (LPS), or entered in contact with a biomaterial. Simulation results were obtained in terms of activation variables and cell counts, and were used to validate the model against experimental results available in literature. These outcomes were consistent with experimental observations, suggesting that the model proposed here is a promising tool that can be further extended to fully describe the whole foreign body reaction (FBR) process.

In the following sections, we first validate the original ODE and AB models [3,8] for macrophages polarization induced by external stimuli, against a set of experimental data. We then describe the AB model encompassing macrophage and fibroblast interactions in the immune response, that is the main novelty of the present work. Finally, we validate also this model against data from the literature as well as exploring few selected scenarios for the evolution of the macrophages/fibroblasts ecosystem. This allows us to discuss limitations of the model that are challenges for its future expansion.

## 2 Materials and methods

### 2.1 Macrophage modeling

For describing the polarization of macrophages under an external stimulus (a pathogen or, indirectly, the presence of a biomaterial), we considered two different modeling approaches. The ODE models introduced in [8] and [3] simulate subcellular pathways with a set of differential equations that describe the change in concentrations of the species involved and their phenotypes. Both models are developed around a pro-inflammatory and a pro-healing module, representing tumor necrosis factor $\alpha$ (TNF-$\alpha$) and interleukin-10 (IL-10) cytokines, respectively. The whole dynamics is mediated by the nuclear factor-$k$B (NF-$k$B).

In [8] IL-10 regulates its own production and TNF-$\alpha$ production through NF-$k$B (in positive and negative feedback, respectively). In addition, TNF-$\alpha$ regulates its own production through NF-$k$B as well, in positive feedback (Fig 1). The system was stimulated with a constant concentration of LPS. We will refer to this model as $Maiti_{ode}$ in the following.

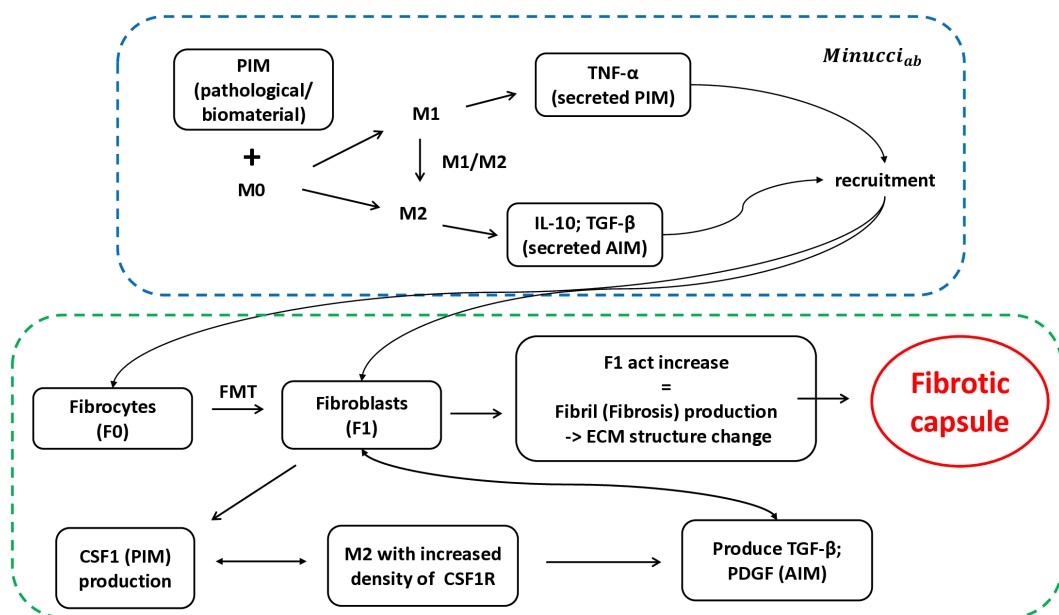

**Fig 1. Schematic of the proposed model.** The sketch includes the stimulation of macrophages by PIM initially administered to the system, which leads to their polarization and cytokine secretion. This module is taken from AB model in the literature [3] (blue dotted box). To this macrophage polarization module, we add the recruitment and subsequent interaction of fibroblasts with M2 macrophages through the CSF1-CSF1R axis. This leads to the increase of the $F_{1act}$ variable, change in the structure of ECM and finally formation of the fibrosis capsule (green dotted box).

In the ODE version of the model in [3], the pro-inflammatory stimulus is represented by supernatant TNF-$\alpha$. Moreover, the Janus kinase 1-tyrosine kinase 2 (JAK1-Tyk2) pathway, that is a relevant step in the activation and translocation of the signal transducer and activator of transcription 3 (STAT3), was added to the IL-10 module. In addition, the suppressors of cytokine signaling (SOCS) 1 and 3 were added to the same module, through more detailed feedback loops. We call this model $Minucci_{ode}$ in the following. On the other hand, the AB scheme in [3] models macrophages as mobile agents, whose interaction is influenced by their spatial position and polarization state through simple rules. Below, we refer to this model as $Minucci_{ab}$.

In the AB model implementation, dimensionless state variables are assumed. Specifically, pro- and anti-inflammatory mediators (PIM and AIM, respectively) act as cytokines that guide macrophage polarization. The resulting activation variables for each phenotype ($M_{1act}$ and $M_{2act}$) regulate PIM and AIM secretion, with SOCS variables controlling their inhibition. Based on activation levels, the macrophage phenotype is assigned to a continuous range of possible states, as follows:

1. if $M_{1act} > 0.5$, then the macrophage is in a pro-inflammatory phenotype ;
2. if $M_{2act} > 0.5$, then the macrophage is in a pro-healing phenotype;
3. if $M_{1act} + M_{2act} > 0.25$, then the macrophage is in an intermediate phenotype between $M_1$ and $M_2$.

$M_{1count}$ and $M_{2count}$ are the counting variables that allow us to monitor the dynamic evolution of the phenotypes.

Initial conditions include, among others, the number of resident macrophages in naive state, randomly distributed over the bidimensional grid, and the PIM located at the center of

the lattice, which diffuse isotropically reaching the simulation grid borders. At each step, the viability of the cell is evaluated according to its assumed lifetime by comparing the actual cell age with the stochastic sampling of the exponential distribution with the lifetime assigned for each phenotype. If alive, each macrophage may randomly diffuse and take up, if empty, one of the eight nearest patches surrounding it. This results in a stochastic diffusion with an excluded volume effect. Based on the position assumed by the cell and on the local and global concentrations of PIM and AIM, activation variables are updated. Next, the macrophage phenotype is adjusted based on these values and the age of macrophages is increased. The mediators (PIM and AIM) uniformly diffuse at each time step over a $3 \times 3$ nearest neighboring pixel grid and decay with their intrinsic lifetime. Finally, the recruitment of new macrophages is implemented (when necessary) based on the total amount of mediators over the simulation lattice. All the variables are consequently updated and another iteration starts, until the total simulation duration is reached. The time step $\tau$ of the simulation, representing the duration of each iteration, was set to 20 minutes, to obtain an appropriate description of the activation dynamics of macrophages.

Both ODE and AB models were originally implemented in MATLAB (MathWorks, Inc.).

**2.1.1 Model validation.** The validation of the models $Maiti_{ode}$ [8] and $Minucci_{ode}$ [3], and of the $Minucci_{ab}$ approach introduced in [3], was performed against experimental data based on fluorescence-activated cell sorting (FACS) analysis on mice. Murine macrophages were incubated for 10 min, 30 min, 1 h, 3 h, 6 h and 12 h with LPS (1 $\mu$g/mL), then washed from LPS and incubated with antibodies for three different pro-inflammatory cytokines: TNF-$\alpha$, interleukin-12 (IL-12) and interferon-gamma (INF-$\gamma$). For each incubation time, gates of cells were created to select those expressing the specific cytokine addressed by the labeled antibody and to remove debris and cell doublets. The resulting percentage of cells in the gate with respect to the total number of cells was used in our validation against the values obtained by the simulation algorithms. The FACS analysis provides cytoplasmatic concentrations of the selected citokines. Therefore, we focused on the kinetics of production of the corresponding species of TNF-$\alpha$ (TNF-$\alpha_{cyto}$) as predicted by the ODE models. In the case of the AB model, we considered the $M_1$ activation and counting state variables ($M_{1act}$ and $M_{1count}$, respectively), which were compared to the three pro-inflammatory cytokines monitored experimentally. The two ODE models were stimulated with the same LPS concentration as in the experiments (1 $\mu$g/mL). In the AB model, we set the pro-inflammatory stimulus at PIM = 30 (adimensional units). Sensitivity analysis and influence of different experimental conditions are further analyzed in S1 File. Specifically, S1 Fig shows the effect of continuous inflammatory stimulation vs early washout, S2 Fig shows the sensitivity analysis of the ODE model, while S3 Fig analyzes the effect of varying 3 rate constants in the ABM model.

## 2.2 The MFF model: Macrophage-fibrocyte-fibroblast crosstalk including pathological PIM

All the models [3,8] reviewed above are limited to the description of the evolution of the phenotype of macrophages. Here, we extend and improve them by introducing the interaction of these cells with fibrocytes and fibroblasts. The lower part of Fig 1 depicts the proposed relationships among macrophages, fibrocytes and fibroblasts. These are inspired by the action of the colony-stimulating factor 1 (CSF1) [9], that we simulate as a PIM produced by fibroblasts. The corresponding receptor (CSF1R) is expressed by $M_2$ macrophages at increased levels. The $M_2$ macrophages produce in turn transforming growth factor $\beta$ (TGF-$\beta$) and platelet-derived growth factor (PDGF), that we model as AIM. In this new model (hereafter called MFF model, which stands for macrophage-fibrocyte-fibroblast model), sizes and

concentrations are used in adimensional units as in [3], but with a direct cross-check to the literature data.

A further improvement with respect to [3] is represented by the inclusion of cell chemotaxis driven by the gradient of the local cytokines concentration. This mechanism proves to be prevalent if compared to cellular diffusion, which is negligible *in vivo*. The chemotaxis is modeled as a stochastic process. The motion probability is computed over a $3 \times 3$ matrix around the cell and it is linearly dependent on the PIM concentration gradient over one pixel. The proportionality parameter $k$ has been derived from the following equation:

$$P_c = k\frac{\Delta c}{\Delta x},$$ (1)

where $P_c$ represents the chemotaxis probability, $\Delta c$ is the concentration gradient and $\Delta x$ is the cell displacement, coinciding with the dimension of the pixel. We estimated $k$ from Eq 1 using experimental data found in [10], following two different approaches. In both cases, the chemotactic probability was hypothesized to be the dimensionless forward migration index (FMI, as defined in [10]). In the first approach, a dimensional parameter has been derived using experimental values of FMI, $\Delta x$ and $\Delta c$ found in [10]. Afterwards, this has been converted to its adimensional version, using the assumption that the dimension of each pixel is $\Delta x = 10\ \mu m$. In this way, we get $k = 0.02$. In the second approach, the proportionality parameter has been computed directly in adimensional unit. More in detail, we assumed that the value PIM = 30 was equivalent to the concentration LPS = 1 $\mu g$/mL used in the experiment described in Sect 2.1.1, since it led to IL-10 concentrations of the same order of magnitude as growth factor concentrations found in [10]. Therefore, using $\Delta x = 15$ pixels and $\Delta c = 30$, we obtain $k = 0.5$.

In order to quantify the effect of the different $k$ parameter values, we computed the number, $N_c$, of movements between two pixels of all the cells (macrophages expressing all possible phenotypes, fibrocytes and fibroblasts) due to chemotaxis, at each time step of a simulation.

Differently from what done in [3], diffusion of PIM has been set directly from the jump probability from one pixel to another, $P_{D,pim}$, as follows:

$$P_{D,pim} = \frac{\Delta x}{\sqrt{4D_{pim}\tau}}e^{\left(\frac{\Delta x^2}{4D_{pim}\tau}\right)},$$ (2)

For PIM, Eq 2 is used with $\Delta x = 10\ \mu m$, the time step $\tau = 20$ min and the diffusion coefficient values $D_{pim} = 900\ \mu m^2$/min for TNF-$\alpha$, as found in [11,12].

The jump probability for AIM, $P_{D,aim}$, is computed similarly to PIM, just replacing $D_{pim}$ with $D_{aim} = 780\ \mu m^2$/min for TGF-$\beta$ [11,12].

We then introduced the fibroblasts in the AB model as a separate population. Differently from macrophages, only two phenotypes were considered, fibrocytes ($F_0$ phenotype, or non-activated) and fibroblasts ($F_1$ phenotype, or activated). To control the activation state of fibrocytes, and allow their possible differentiation, a fibroblast activation variable $F_{1act}$ was introduced, which steers the cell towards the activated or non-activated state. Counting variables for the two new cell populations were also introduced, similarly to what done for the macrophages. Due to the pro-healing behavior of both cell types (fibrocytes and fibroblasts), the kinematic rates were taken equal to the corresponding parameters for $M_2$ macrophages. Values of the remaining parameters were not changed compared to the model proposed in [3]. The average lifespan for $M_0$ and $F_0$ cells has been set to $24 \pm 6$ hours, while for activated macrophages and fibroblasts we chose $48 \pm 12$ and $336 \pm 84$ hours, respectively [3,10].

Since we want to simulate an *in vivo* scenario, we included the recruitment of cells from other distant compartments of the organism under investigation. We allowed the recruitment only of $M_0$ macrophages. This means that macrophages in $M_1$, $M_2$ and intermediate phenotypes are obtained only by differentiation of macrophages in $M_0$ state (either recruited or resident ones). On the other hand, we allowed for the recruitment of both fibrocytes, $F_0$, and fibroblasts, $F_1$, as it is likely to occur in a real scenario. The recruitment probability for $F0$ and $F1$ cells is described by a second-order Hill function dependent on the local (i.e. in the pixel where the cell is supposed to be recruited) amount of PIM and AIM. The Hill parameters for this activation function are *Patho_PIMRecruitScale_f* and *AIMRecruitScale_f*, for PIM and AIM driven recruitment respectively (see also Table 1, where we specified different values that a parameter assumes depending on the situation, whenever variations occurred). Similarly, the probability to recruit a macrophage to a selected pixel of the grid depends on the total amount of PIM over the entire lattice and on the value of the local AIM in the same pixel, by means as a second-order Hill function with cooperative parameter *AIMRecruitScale* [3] (see also Table 1). This implementation was included to make the movement of macrophages towards the center of the lattice smoother, as experimentally observed in [7].

In general, most of the kinetics reproduced in the MFF model, including cell recruitment, PIM and AIM production, activation variables update, are based on the Hill equation in the following form:

$$\frac{(ax)^n}{(ax)^n + k}, \tag{3}$$

where $x$ is the concentration of PIM or AIM (locally in the single pixel or globally over the entire grid, depending on the law), $a$ is the cooperative parameter, $n$ is the order of the function (Hill coefficient) and $k$ is a scale factor.

**Table 1. List of the novel parameters used in the MFF model proposed in this work, similarly to [3]\*. All of them are in dimensionless units.**

| New parameters | Value | Description |
|---|---|---|
| FibroAntiInflammatoryRate | 0.85 | Rate at which fibrocytes and fibroblasts produce AIM (default configuration) |
| | 0.60 | Rate at which fibrocytes and fibroblasts produce AIM (under pathological stimulus) |
| FibroProInflammatoryRate | 0.35 | Rate at which fibrocytes and fibroblasts produce PIM (default configuration) |
| | 0.55 | Rate at which fibrocytes and fibroblasts produce PIM (under pathological stimulus) |
| FNegativeFeedbackRate | 0.003 | Rate at which $F_{1act}$ activation decays |
| RecruitmentFFTerm ($k$) | 30 | Regulates effectiveness of fibrocytes and fibroblasts recruitment by PIM and AIM (default configuration) |
| | 3 | Regulates effectiveness of fibrocytes and fibroblasts recruitment by PIM and AIM (with biomaterial) |
| F1ActScalar | 0.065 | Rate at which increase $F_{1act}$ is increased by AIM |
| F1ActHillParameter ($k$) | 0.85 | Regulates effectiveness of increasing $F_{1act}$ via AIM |
| F1SOCSInfinity | 7 | Regulates effectiveness of inhibiting $F_{1act}$ by SOCS |
| Patho_PIMRecruitScale_f ($a$) | 0.5 | Rate at which pathological PIM recruit fibrocytes and fibroblasts |
| Fix_PIMRecruitScale_f ($a$) | 0.2 | Rate at which fixed PIM (when included) recruit fibrocytes and fibroblasts |
| Fix_PIMRecruitScale_m ($a$) | 0.7 | Rate at which fixed PIM (when included) recruit macrophages |
| AIMRecruitScale_f ($a$) | 1 | Rate at which AIM recruit fibrocytes and fibroblasts |

\* Parameters for the macrophages are listed in this reference paper.
($a$) and ($k$) refer to parameters $a$ and $k$, respectively, in Eq (3).

**2.2.1 Validation of the MFF model under pathological stimulus.** The MFF model proposed in Sect 2.2 was quantitatively validated against experimental data from literature. The average secreted PIM and AIM resulting from simulations were compared to the average mRNA expressions of all the pro- and anti-inflammatory cytokines, respectively, reported for cultures of macrophages in [13] (see data in Fig 2, ref [13]). Simulated trends have been computed over $N = 25$ runs, to approximate the number $n \geq 5$ of experiments over 5 donors as reported in [13]. More in detail, the model was stimulated with a center-peaked Gaussian distributed PIM with maximum value 1500 and AIM with maximum value 58.9 so as to reproduce the concentrations of LPS ([$LPS$] = 50$\mu$g/mL) and dexamethasone ([$Dex$] = $5 \times 10^{-6}$ M), respectively, using the assumption that 1 $\mu$g/mL of LPS corresponds to PIM = 30. We also introduced 1280 resident macrophages at $M_0$ state at the beginning of simulations, corresponding to the 80 % confluence, as indicated in [13]. Moreover, for validation purpose, we assumed that newly recruited cells are located along a circular crown encompassing the central stimulus, and then move under chemotaxis effect. The remaining values of the variables and parameters of the model are the same as in [3] and in Table 1, except for the two rates regulating production of PIM and AIM by fibroblasts, that are now changed to 0.55 and to 0.60, respectively.

At the end of simulation time, at $t = 6$ days, the average kinetics of total PIM and AIM secreted was collected among the 25 runs and the time average was computed for the analysis. For both PIM and AIM dynamics, the analysis was conducted considering the mean values.

Twenty-five more simulations, each lasting 6 days, were run following the same approach, but with initial PIM = AIM = 0, thus simulating the control situation. We finally performed the one-way analysis of variance (ANOVA) among kinetics of mediators, followed by a Bonferroni test, to explore the difference between treatment and no treatment scenarios in simulations, as done in [13].

## 2.3 The MFF model: Macrophage-fibrocyte-fibroblast crosstalk including biomaterial

In order to simulate an implanted biomaterial, a complementary inflammatory situation was considered. A permanent concentration of PIM is assumed over the central portion of the simulation space. We call this condition "mechanical" stimulus, as opposed to the "chemical" one. The latter simulates in vitro administered chemical stimulus (like LPS) or the presence of a pathogen that can diffuse in the organism.

This mechanical PIM is located at the center of the grid and described by a constant value of PIM distributed over a connected area. The PIM neither decays nor diffuses in the grid, and it is not involved in the chemotaxis phenomenon, but triggers the FBR, thus leading to the secretion of pro- and anti-inflammatory cytokines (pathological PIM and AIM, respectively).

The recruitment probability of fibrocytes ($F_0$) and fibroblasts ($F_1$) have been modeled as a Hill function that depends on the PIM and AIM values in the pixel to which these cells are recruited. The corresponding Hill cooperative parameters are *Patho_PIMRecruitScale_f* and *AIMRecruitScale_f*, respectively. They are appropriately chosen to reproduce experimental results and reported in Table 1. The remaining parameters introduced as rates for the novel interactions in the MFF model are listed in Table 1.

Fig 1 shows a schematic representation of the cells, biochemical signals and pathways involved in the MFF model, which lead to the formation of the fibrotic capsule. In the same scheme, both pathological and mechanical PIM have been considered. The initial AB model that includes only macrophages is depicted in the upper part of the figure for a direct comparison to the model developed here (lower part of Fig 1), enriched with the interaction among

fibrocytes and fibroblasts and macrophages. The transition from $F_0$ to $F_1$ depends on the level of $F_{1act}$ variable, whose increase indicates the beginning of the fibrosis production and therefore a denser and stiffer ECM. Fibroblasts produce the CSF1 factor, simulated as PIM, and interact with $M_2$ macrophages expressing enhanced level of CSF1R receptor. As a consequence, pro-healing macrophages will produce growth factors, such as TGF-$\beta$ and PDGF, which have been modeled as AIM and will increase the level of $F_{1act}$ variable, finally leading to the formation of the fibrotic capsule.

**2.3.1 Validation of the MFF model with biomaterial.** For the validation of the MFF model stimulated by the presence of a biomaterial ("mechanical PIM") introduced in Sect 2.3, we used experimental results presented in [7], where an implant was placed in the subcutaneous space on the back of five rats and the average number of macrophages (without specifying phenotypes) and fibroblasts was collected over seven and twenty-eight days at five different distances from the implant. In order to reproduce this experimental scenario, average counting variables of macrophages and fibroblasts were collected in five simulations, over seven and twenty-eight days, at three different distances from the fixed PIM modeled as a 14 × 14 pixel square located exactly at the center of the lattice. The decision to count cells at only three distances (compared to the five observation regions used in [7]) is driven by the limited space available in the simulation volume compared to the real-world scenario. More in detail, data were collected in three squares 3×3 pixels wide, distributed equally over a line of 13 pixels on each side of the biomaterial and are $\Delta x = 10\ \mu m$ apart from each other. In order to enhance the reliability of cell quantification, macrophages and fibroblasts were counted on all four sides relative to the central implant, resulting in a total of twelve areas (four at each specified distance). Fig 2 shows the sketch graph for the data collection geometry, similar to what is reported in [7] (Fig 2).

In order to better reproduce a realistic scenario, the MFF AB model was stimulated with a mechanical stimulus of $PIM = 20$, assuming no resident macrophages in $M_0$ state at the beginning of simulations. The transition from the tissue to the biomaterial is smoothed by convolving its volume with a Gaussian smoothing kernel with standard deviation of 2.

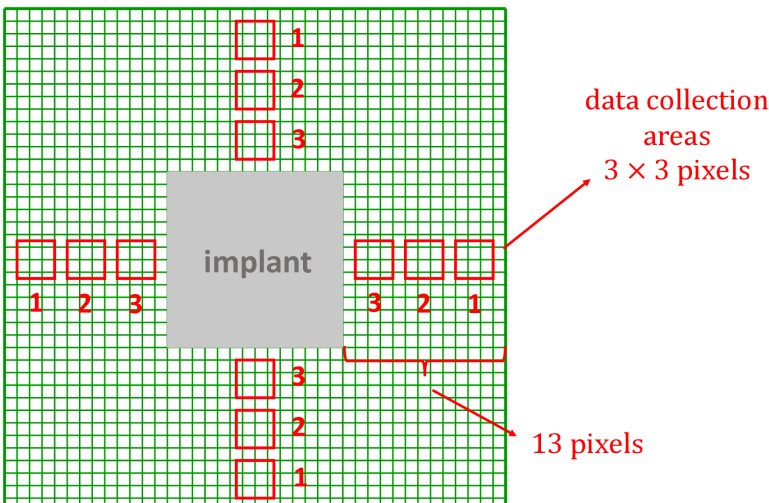

**Fig 2. Schematic representation of the biomaterial and data collection areas.** The implant is represented as a 14 × 14 pixel square (full grey) at the center of the 40 × 40 pixel grid (green), whereas the selected areas for counting cells are depicted as 3 × 3 pixel squares (red lines) surrounding the central implant at its four sides.

This smoothing is thought to represent the presence of adsorbed plasma proteins (like in opsonization) on the surface of the biomaterial.

Newly recruited cells are located along a circular crown encompassing the central biomaterial. The values of the simulation parameters are the same as those used in Sect 2.2.1 for the validation of the MFF model under pathological stimulus. The only exceptions are given by the two parameters that regulate the effectiveness of macrophage and fibroblast recruitment activated by PIM and AIM, referred to as *RecruitmentMMTerm* and *RecruitmentFFTerm*, respectively, which appear as scale factors in the corresponding Hill equations (see Eq 3) and are set to 3 (instead of 30).

Finally, we compared the spatial distribution of the number of macrophages and fibroblasts between the simulation and the experimental results observed in [7], providing mean values and standard deviations (SD) for each case.

## 3 Results

### 3.1 Experimental validation of the pro-inflammatory dynamics of macrophages

To validate the models introduced in [8] and in [3], as described in Sect 2.1.1, simulations were run for 12 hours and compared to *in vitro* data obtained from FACS analysis. For the ODE approaches, cells were stimulated with LPS = 1 $\mu$g/mL. For the AB approach, we assumed an equivalent PIM = 30, distributed over a $13 \times 13$ bidimensional grid at the center of our $40 \times 40$ pixel volume. In addition, to assess stochasticity in the AB model, $N = 30$ simulations were run, each with a time step of 20 minutes, and the average values of each state variable were considered. In this case, at the beginning of simulations, 1280 non-activated (or at $M_0$ state) macrophages were distributed randomly over the simulation space (where each macrophage can occupy one pixel at a time), so as to approximate the 80 % confluence. The recruitment process was not included at this stage, to better reproduce *in vitro* experiment conditions.

In order to quantify the agreement of the simulated dynamics with the available experiments, the Pearson's correlation coefficient $\rho$ was used. Moreover, to be able to compare the results of the two ODEs models, the time course of TNF-$\alpha_{cyto}$ was normalized with respect to its maximum value and sampled at time points closest to the six experimental measurement times (10 min, 30 min, 1 h, 3 h, 6 h and 12 h) . In case of AB model, we computed the average of $M_{1act}$ and $M_{1count}$ produced at each run time, sampling them as for ODE approaches. Finally, in order to compare the ODEs results to the findings from FACS data, only experimental data on the TNF-$\alpha$ expression was considered. In the case of the AB model, instead, we computed the correlation of the activation and count variables with all the available experimental cytokines.

Fig 3 shows the percentage of the cytokine's production predicted by the ODE and AB models (where the former were computed with 1 $\mu$g/mL of LPS, as in the experimental setup), as well as the values obtained from FACS experiments for the three different cytokines monitored (TNF-$\alpha$, IL-12 and INF-$\gamma$). These represent technical replicates, obtained from repeated measurements of the same cell culture under identical experimental conditions. Specifically, for each sample, we performed three independent FACS measurements. The technical uncertainty is minimal (maximum variation ∼1%), indicating high measurement precision. In contrast, when performing biological triplicates, we observed a higher intrinsic variability (∼5–10%), as expected.

Table 2 shows $\rho$ values between each pair of variables considered in this first study. We found the best agreement between simulations and experimental results when comparing

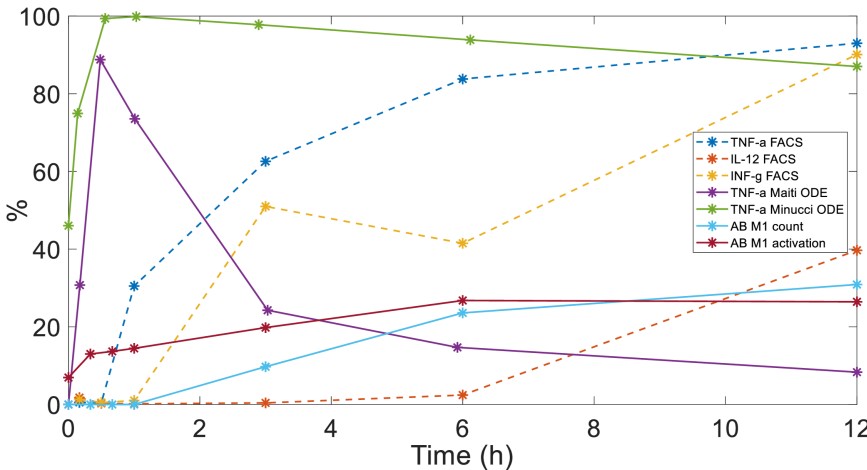

**Fig 3. Correlation analysis between model (ODE and AB) and experiment (FACS) results.** All the simulation variables and experimental measurements were reported in percentage of cells, coming from normalization and gating, respectively. To highlight that only six values from technical triplicates (maximum variation ~1%), in the FACS analysis were collected and used, respective curves have been dotted.

**Table 2. Pearson's correlation coefficient $\rho$ between M1 variables from ODE and AB models (TNF-$\alpha_{cyto}$, $M1_{act}$ and $M1_{count}$, respectively) and pro-inflammatory cytokines from FACS (TNF-$\alpha$, IL-12, INF-$\gamma$).**

| Model | model variable | FACS data | $\rho$ | $p$-value |
|---|---|---|---|---|
| $Maiti_{ode}$ | TNF-$\alpha_{cyto}$ | TNF-$\alpha$ | −0.75 | 0.09 |
| $Minucci_{ode}$ | TNF-$\alpha_{cyto}$ | TNF-$\alpha$ | 0.13 | 0.81 |
| $Minucci_{ab}$ | $M1_{act}$ | TNF-$\alpha$ | **0.96** | 2.2e-3 |
| | $M1_{act}$ | IL-12 | 0.60 | 0.21 |
| | $M1_{act}$ | INF-$\gamma$ | 0.87 | 0.02 |
| | $M1_{count}$ | TNF-$\alpha$ | **0.93** | 0.01 |
| | $M1_{count}$ | IL-12 | 0.76 | 0.08 |
| | $M1_{count}$ | INF-$\gamma$ | 0.92 | 0.01 |

the $M1_{act}$ and $M1_{count}$ variables produced by the AB model with TNF-$\alpha$ ($\rho$ = 0.96 and 0.93, respectively, $p$-value < 0.05). On the other side, TNF-$\alpha_{cyto}$ value from the ODE model proposed in [8] show a negative correlation with experimental data ($\rho$ = −0.75, without statistical significance), while $\rho$ = 0.13, (without statistical significance) for the ODE version in [3].

These results do not seem to be due to a particular choice of the simulation parameters. We investigated the dependence of the activation $M_{1act}$ and $M_{2act}$ state variables and the cell counts ($M_{1count}$ and $M_{2count}$), produced by running the AB model, on the pro-inflammatory stimulus and on the initial number of macrophages in $M_0$ state. Results of this analysis, summarized in Fig 4, reveal that at 12 hours, higher levels of PIM correspond to larger values of $M_{1act}$ and $M_{1count}$, even with few initial $M_0$ macrophages. On the other hand, the amount of $M_{2act}$ and the number of macrophages in $M_2$ depend on the initial number of macrophages in $M_0$ more than on the pro-inflammatory stimulus provided. This behavior is expected since the larger the number of macrophages, the more cytokines will be overall secreted, which will thus increase their concentration: macrophages are consequently recruited and a feedback loop is set.

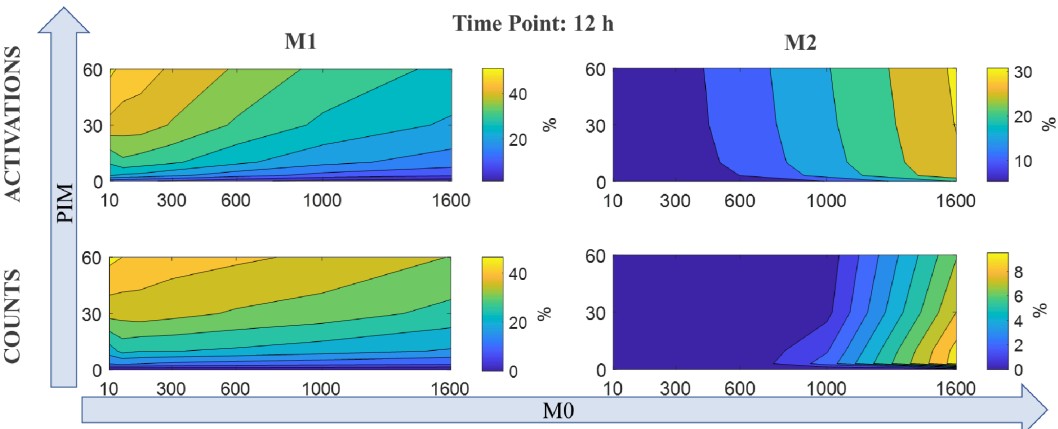

**Fig 4. AB model state variable dependence.** Contour plots showing dependence of $M_1$ and $M_2$ state variable (top row: activation, bottom row: count) on the initial number of non-activated macrophages $M_0$ (horizontal axis) and on the initial value of PIM provided (vertical axis), at the simulation time of 12 hours.

## 3.2 Experimental validation of the macrophage-fibroblast model under pathological stimulus

In Fig 5 we report the values for the secreted PIM and AIM cytokines, averaged over twenty-five simulations (open, slanted hatched bars). Three cases were considered: a control case

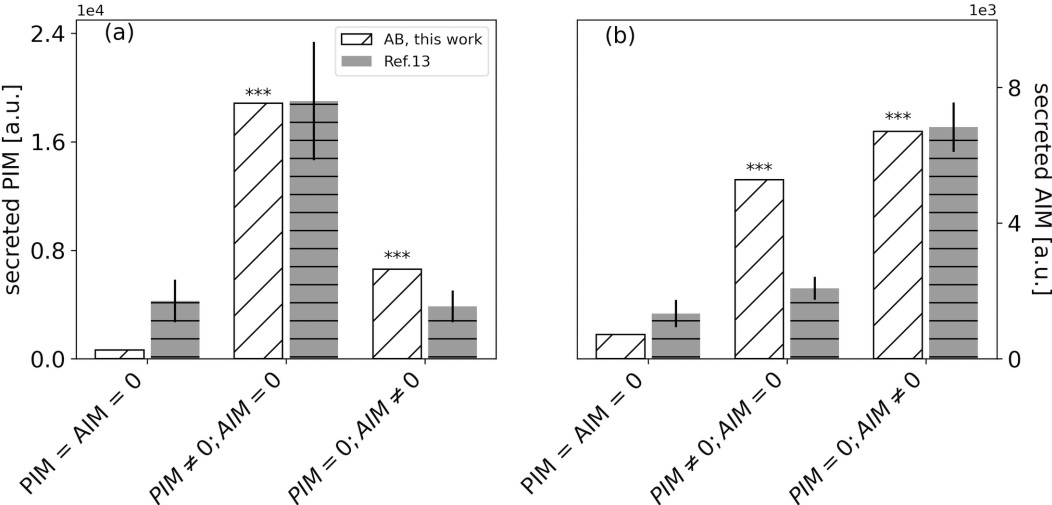

**Fig 5. Total PIM and AIM secreted in three different stimuli scenarios compared to experimental data.** The slanted white bars ("AB, this work") report the mean values (among the twenty-five runs) of secreted PIM (panel a) and AIM (panel b) cytokines for (1) a control case (labeled as *"PIM = AIM = 0"*), (2) a Gaussian distribution of PIM peaked at 1500 (PIM ≃ 1500, labeled as *"PIM ≠ 0; AIM = 0"*) and (3) for stimulation with AIM peaked at 58.9 (AIM ≃ 58.9, labeled as *"PIM = 0; AIM ≠ 0"*). The PIM and AIM values are reported in adimensional unit (a.u.). The gray horizontally hatched bars report the data taken from ref [13] (Fig 2, therein). These data were normalized to their maximum cytokine expression. *** $p$-value < 0.001. For comparison with our AB results, the values of the PIM cytokines were the averages of the TNF-$\alpha$ and matrix metalloproteinase (MMP7) data (Fig 2 in [13]). The values of the AIM cytokines were averaged of the PDGF-A, PDGF-B, TGF-$\beta$ and alternative macrophage activation-associated CC-chemokine-1 (AMAC-1) data reported in Fig 2 of ref [13]. For the evaluation of the PIM and AIM stimuli, we averaged the columns LPS and interferon (IFN), and the columns IL-4 and *Dex*, respectively.

(AIM = PIM = 0), a system stimulated by adding PIM (PIM ≠ 0; AIM = 0) and a system stimulated by adding AIM (PIM = 0; AIM ≠0). The cytokines were described by Gaussian spatial distributions with peak values $PIM_{max} = 1500$ and $AIM_{max} = 58.9$. All the concentrations are reported in arbitrary units.

We found a significant difference in PIM secretion between the control and the stimulation conditions. A significant difference is also found between the two stimulation conditions ($p$ value < 0.001 for each pairwise comparison considered). Similar results are observed for the secreted AIM. When no stimulus is applied, we obtain comparable levels of secreted PIM and AIM (PIM ≈ 150; AIM ≈ 220). In addition, as expected, when stimulated with PIM, the system produces more PIM than AIM (PIM $\approx 1.9 \times 10^4$ vs. AIM $\approx= 5300$). This trend is observed when AIM is provided as input as well, but to a less extent if compared to the previous scenario (PIM = 6600 vs. AIM = 6700).

The results of the MFF simulation model compare quantitatively also with experimental data. In Fig 5, the data taken from Fig 2 of ref [13] were reported after normalization to the maximum value of the secreted cytokine (gray, horizontally hatched bars). The general trend is recovered by the simulations with great accuracy, especially for the case of PIM secretion. For the case of secreted AIM cytokines, the MFF model predicts, with the chosen set of parameters, a larger increase of cytokines under PIM stimulation than was experimentally observed. In any case, both the model and the data indicate that there is some level of crosstalk between PIM and AIM pathways that results in the increase of AIM secretion under PIM stimulus (panel b, *PIM* = 0; *AIM* ≠ 0) with respect to the control.

## 3.3 Experimental validation of the macrophage-fibroblast model under mechanical stimulus

Our aim here is to reproduce the experiments reported by Yang et al. [7] about the distribution of macrophages and fibroblasts at increasing distance from a biomaterial. We performed this by simulating the reaction to a biomaterial as reported in Figs 1 and 2.

Fig 6 shows the total number of macrophages and fibroblasts resulting from counting in three areas (area 1: farthest from the implant; area 3: nearest to the implant), relative to the four sides of the central square, after 7 days and 28 days, for five different simulations. The average curve is also shown to guide the eye. The number of fibroblasts increases by about 60% on the average from 7 to 28 days in the observation regions 1 and 2. The number of macrophages decreases on average as the observation region gets closer to the biomaterial and decreases by more than a factor of 2 from day 7 to day 28. Both findings are consistent with experimental results reported in [7] (see insets in Fig 6).

At 28 days, the concentration of fibroblasts in the region of interest closest to the biomaterial is lower than the average concentration observed in areas farther from it (panel d in Fig 6). This outcome, which partially diverges from the experimental findings in [7], can be attributed to the fact that, in long-term implants, many cells are located within the same spatial regions as the biomaterial. Moreover, lower intersimulation variability is observed for the fibroblast number at longer observation times, as also found in the experimental results reported in ref [7]. As reported in Table 3, SD $\simeq$ 2.5 for day 7 and SD $\simeq$ 2 for day 28 in the first two observation regions. On the other hand, we found larger variability among the simulated curves of the macrophages spatial distribution than reported in [7]. This may be due to the inclusion in the cell counting of all available states, pro-inflammatory and pro-healing

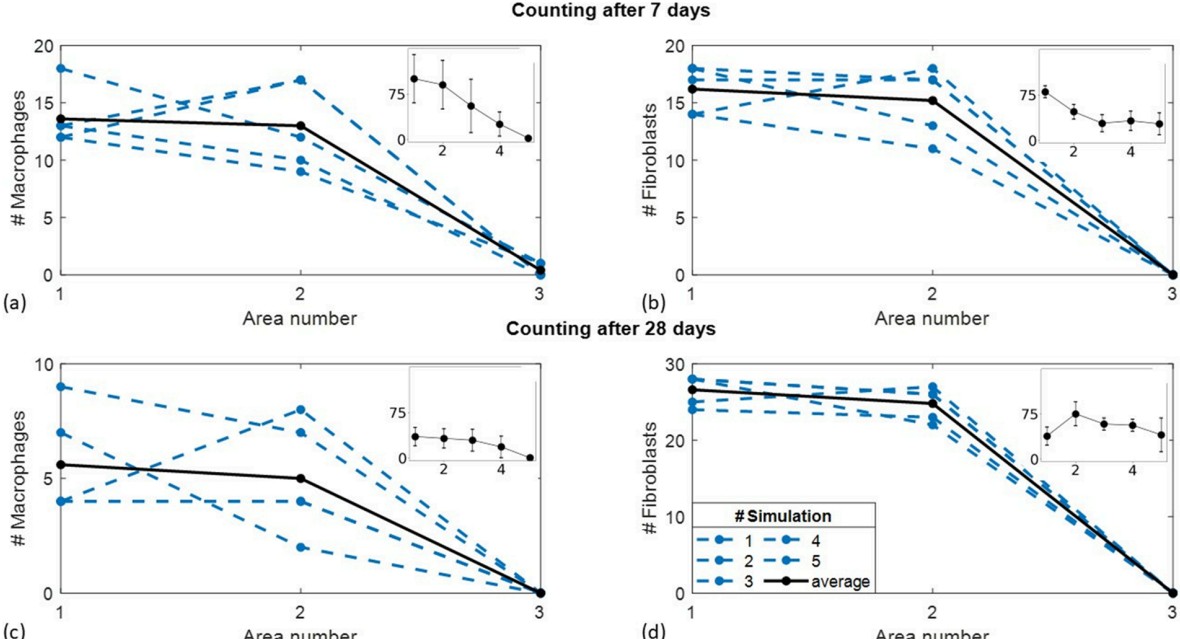

**Fig 6. Cell counting in the presence of the implant as a function of the distance from the implant site.** Total number of macrophages (left) and fibroblasts (right) counted on 7th (a and b) and 28th day (c and d) within the three collection areas (see Fig 2), for each of the five simulations considered. The average trend is reported to guide the eye. The insets in all the four panels report the trend measured in [7] for the corresponding type of cells and incubation times.

**Table 3. Summed counts of macrophages (M) and fibroblasts (F) for each area type (1: furthest, 3: nearest), at five simulations lasting 7 or 28 days. For both simulation sets, average and standard deviation values are also reported.**

| Area | 1 | | 2 | | 3 | |
|------|-----|-----|-----|-----|-----|-----|
| Simulation | M | F | M | F | M | F |
| *7 days* | | | | | | |
| #1 | 12 | 14 | 17 | 11 | 0 | 0 |
| #2 | 12 | 18 | 9 | 17 | 1 | 0 |
| #3 | 13 | 18 | 17 | 13 | 0 | 0 |
| #4 | 18 | 14 | 12 | 18 | 1 | 0 |
| #5 | 13 | 17 | 10 | 17 | 0 | 0 |
| Average | 13.6 | 16.2 | 13.0 | 15.2 | 0.40 | 0 |
| SD | 2.50 | 2.05 | 3.80 | 3.03 | 0.55 | 0 |
| *28 days* | | | | | | |
| #1 | 4 | 28 | 4 | 26 | 0 | 0 |
| #2 | 7 | 25 | 2 | 27 | 0 | 0 |
| #3 | 9 | 24 | 7 | 23 | 0 | 0 |
| #4 | 4 | 28 | 8 | 22 | 0 | 0 |
| #5 | 4 | 28 | 4 | 26 | 0 | 0 |
| Average | 5.60 | 26.6 | 5.00 | 24.8 | 0 | 0 |
| SD | 2.30 | 1.95 | 2.45 | 2.17 | 0 | 0 |

phenotypes, which behave differently in time and space. This also explains the decrease of the total number of macrophages from 7 to 28 days (from about 13.5 to about 5.5 for for areas 1 and 2).

### 3.4 Dynamics of the macrophage-fibrocyte-fibroblast model and simulation scenarios

In the previous sections, we have assessed to what extent the proposed AB model can reproduce *in vitro* and *in vivo* experimental data in mice. We now proceed to a wider exploration of the predictive potential of the MFF model by investigating four scenarios.

In order to study the interaction among different types of cells, for each of the two different inflammatory stimulus configurations considered in Sects 2.2 and 2.3 (pathological or mechanical PIM, respectively), two simulation scenarios were considered, which differ from each other by the presence of resident macrophages in the tissue. The choice not to include $M_0$ macrophages was made to reproduce the experimental conditions in [7], where only macrophages recruited from surrounding tissue were counted. A value of $k = 0.5$ was selected for the proportionality parameter that regulates chemotaxis. We coded the motility of citokines as detailed in the Sect 2.2: the PIM and AIM cytokines diffusion probability was computed by means of Eq 2, finding values $P_{D,pim} \approx 0.021$ and $P_{D,aim} \approx 0.022$ for the eight nearest neighbor pixels in a 3×3 matrix, and $P_{D,pim} \approx 0.833$ and $P_{D,aim} \approx 0.826$ for the central one. All simulations were run for a duration of 36 and 120 hours, simulating a prompt and a chronic response, respectively. The same parameters and initial values of variables were used for each scenario, unless otherwise specified.

For those scenarios under pathological PIM, the stimulus is modeled as in [3], but is smoothed by convolving its volume with a Gaussian smoothing kernel with standard deviation of 2. In case of mechanical PIM, this is located at the center of the grid and described by a constant value of *PIM* = 2 distributed over a connected area.

We summarize the simulation output of each scenario in a set of figures (Figs 7–10), including the PIM distribution (chemical or mechanical depending on the scenario) at $t = 0$ s (panel a), the spatial distribution and kinetics of the various types of cells, for runs of duration $t = 36$ h (panels b) and $t = 120$ h (panels c). In both panels (b) and (c) of each figure, on the left we report the distribution of the cell populations involved, coded in colors (black: empty, white: M0, magenta: M1, blue: M2, yellow: intermediate macrophage state, red: F0, brown: F1), at the last frame of a single representative simulation. Slanted black lines on the colorbar are used to space different entries. Complementarily, in the right panel, the time evolution of the overall number of cells in the same simulation is shown, with the same color code as in the left panel (grey: M0, pink: M1, violet: M2, yellow: intermediate macrophage state, red: F0, brown: F1).

**3.4.1 Scenario 1.A: pathological PIM and no resident macrophages.** In this first simulation scenario, macrophages are assumed to be recruited only from outside the tissue compartments infected by PIM (that reproduces LPS). Therefore, there are no resident macrophages of any phenotype.

In these simulations (see Fig 7), we observe the recruitment of cells ($M_0$, $F_0$ and $F_1$) mainly towards the source of inflammation (center of the lattice), since it strongly depends on the PIM. In the meanwhile the state variables $M_{1act}$ and $M_{2act}$ are updated at each pixel, depending on the level of the cytokines, and the macrophages differentiate by changing their phenotype according to the rules given in Sect 2.1. We also find a prompt recruitment of $F_0$ and $F_1$ cells, which are directly attracted by the pathological PIM. At $t = 36$ h, a provisional capsule begins to appear, mainly formed by $M_2$ and $F_1$, and these types of cells remain predominant until the end of the simulation time $t = 120$ h, as expected. Once 120 hours have passed, fibroblasts, $M_2$ macrophages and fibrocytes form the fibrotic capsule at the center of the lattice, and their density depends on the PIM recruitment rate coefficient

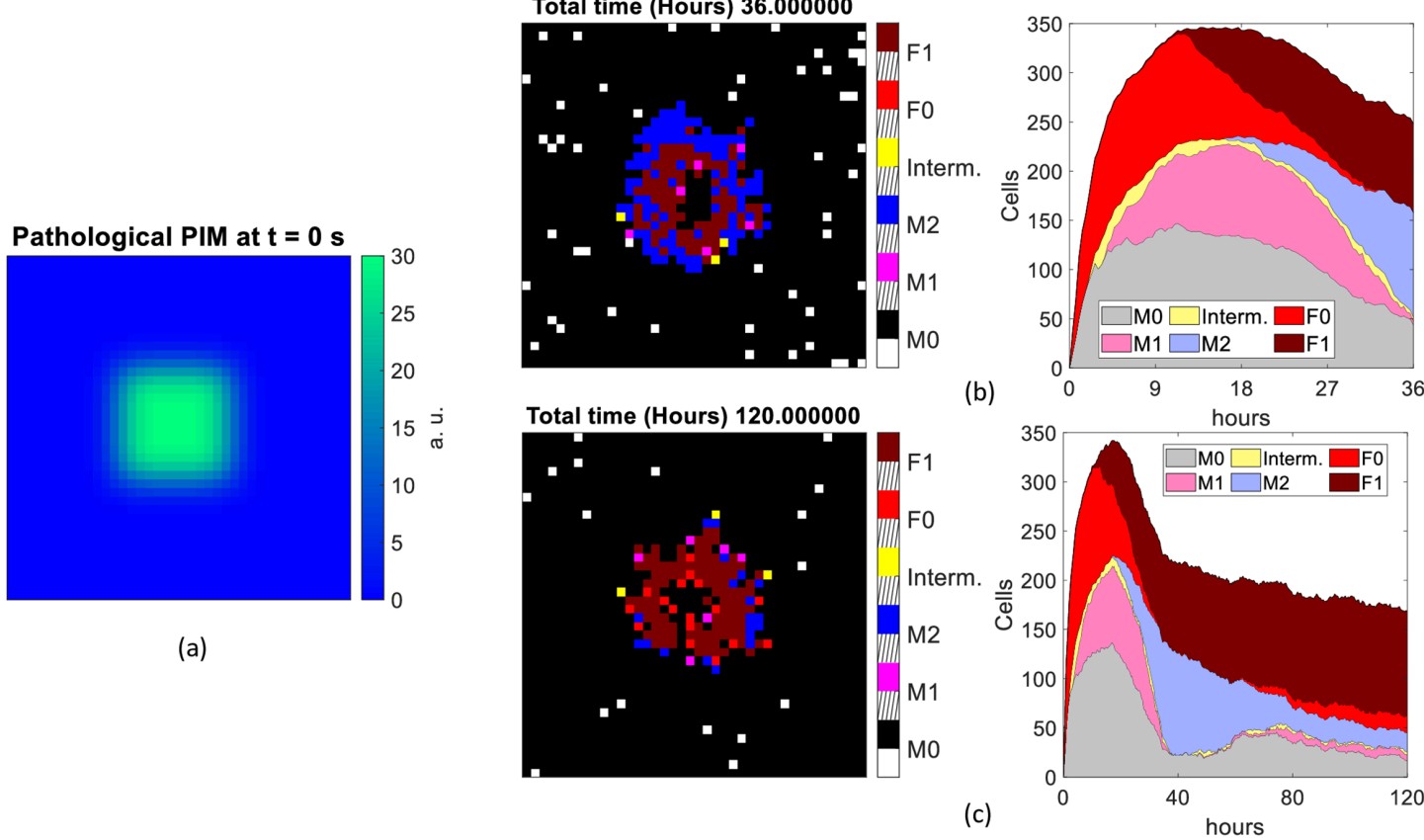

**Fig 7. Scenario 1A. Cell population distributions assuming no resident macrophages and a pathological stimulus as input.** Initial distribution of pathological PIM used (a). The single last frame (left side) and the overall number of cells counted at each time of the simulation (right side) lasting 36 hours (b) and 120 hours (c).

*Patho_PIMRecruitScale_f* and on the AIM recruitment rate coefficient *AIMRecruitScale_f*, that are set at 0.5 and at 1, respectively (see Table 1).

**3.4.2 Scenario 1.B: Pathological PIM and 400 resident macrophages.** The alternative scenario that we considered includes 400 non-activated macrophages at the beginning of the simulation, corresponding to tissue-resident macrophages. This number was chosen to simulate the 25 % cell confluency in a $40 \times 40$ pixel grid.

In this case (see Fig 8), after 36 hours we observe a denser provisional capsule in the center of the grid than in the scenario 1A, likely due to the increased initial number of macrophages. It is predominantly made up of $M_2$ and $F_1$ cells, which form a layer in the bidimensional grid. The surrounding pixels are mainly occupied by macrophages in $M_0$ and intermediate phenotype. Differently from scenario 1.A, the recruitment of fibrocytes ($F_0$) is limited to the first ten hours approximately, whereas all the remaining cells appear until the end of the 36h simulation. The percentage of cells in the intermediate state is also remarkably higher than in scenario 1.A, as well as the $M_2$ macrophages, particularly at the end of the prompt inflammation phase. Starting from 80 hours of system evolution, we mostly observe fibroblasts, fibrocytes, $M_0$ (for the recruitment process) and pro-healing macrophages, as expected, forming the fibrosis capsule. Overall, this scenario indicates the presence of a larger reaction and a faster transition to the chronic state than in the 1A scenario.

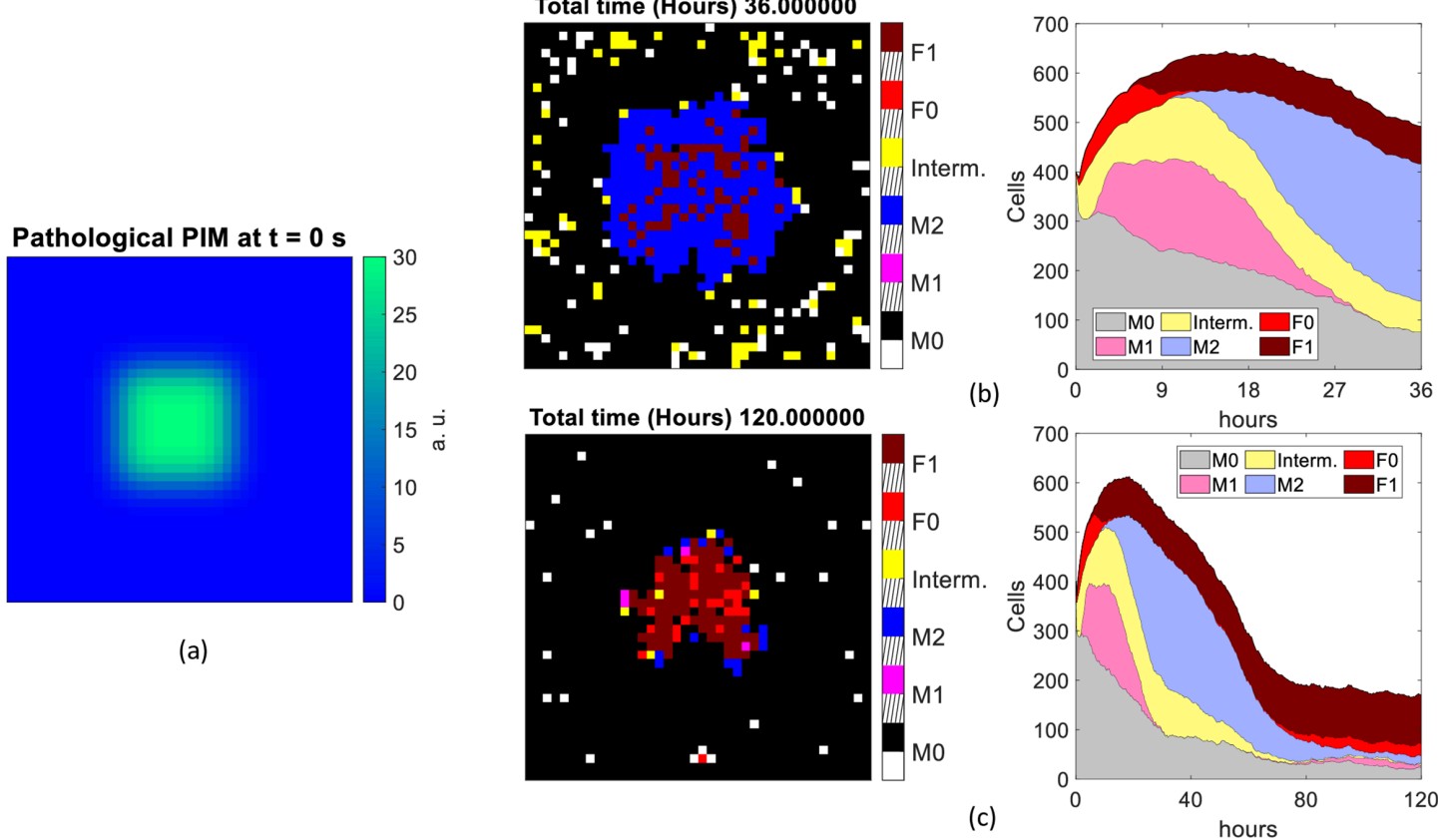

**Fig 8. Scenario 1B. Cell population distributions assuming 400 resident macrophages and a pathological stimulus as input.** Initial distribution of pathological PIM used (a). The single last frame (left side) and the overall number of cells counted at each time of the simulation (right side) lasting 36 hours (b) and 120 hours (c).

**3.4.3 Scenario 2.A: Mechanical PIM and no resident macrophages.** We now move to study a scenario more closely related to FBR. We insert a biomaterial as a constant PIM (referred to as mechanical PIM), located in the center of the lattice in the same position as the pathological PIM used in the previous scenarios. Differently from case 1.A and 1.B, it does not undergo diffusion or decay, but triggers the secretion of circulating PIM simulating pro-inflammatory cytokines, as explained in Sect 2.3.

Both at earlier and later times, we observe a more localized capsule covering the biomaterial, for the different nature of the PIM (fixed stimulus). It is mainly formed by $M_1$, $M_2$ and $F_1$ (see Fig 9 (b) and (c), left side). There is a marked difference with the case of the chemical stimulus. Namely, the space is denser in $M_1$ macrophages in the prompt phase and it is mainly composed of $M_2$ macrophages at 120 hours. In terms of immune response kinetics, we find relevant changes with respect to cases 1.A and 1.B. (1) The cell number increases smoothly over the whole time stretch. (2) It does not show any prompt response followed by a decrease of the number of immune cells for simulation times larger than 30 hours. (3) We find a larger variability in cell distributions than in the previous scenarios. For the first 36 hours, pixels are mainly occupied by macrophages expressing $M_1$ phenotype. Afterwards, pro-healing macrophages ($M_2$) and fibroblasts ($F_1$) dominate (see Fig 9 (b) and (c), right side). Over even longer simulation times (up to ten days), the former ($M_2$) reach a plateau, the latter ($F_1$) increase their density. This is likely originated by a delayed response with respect to

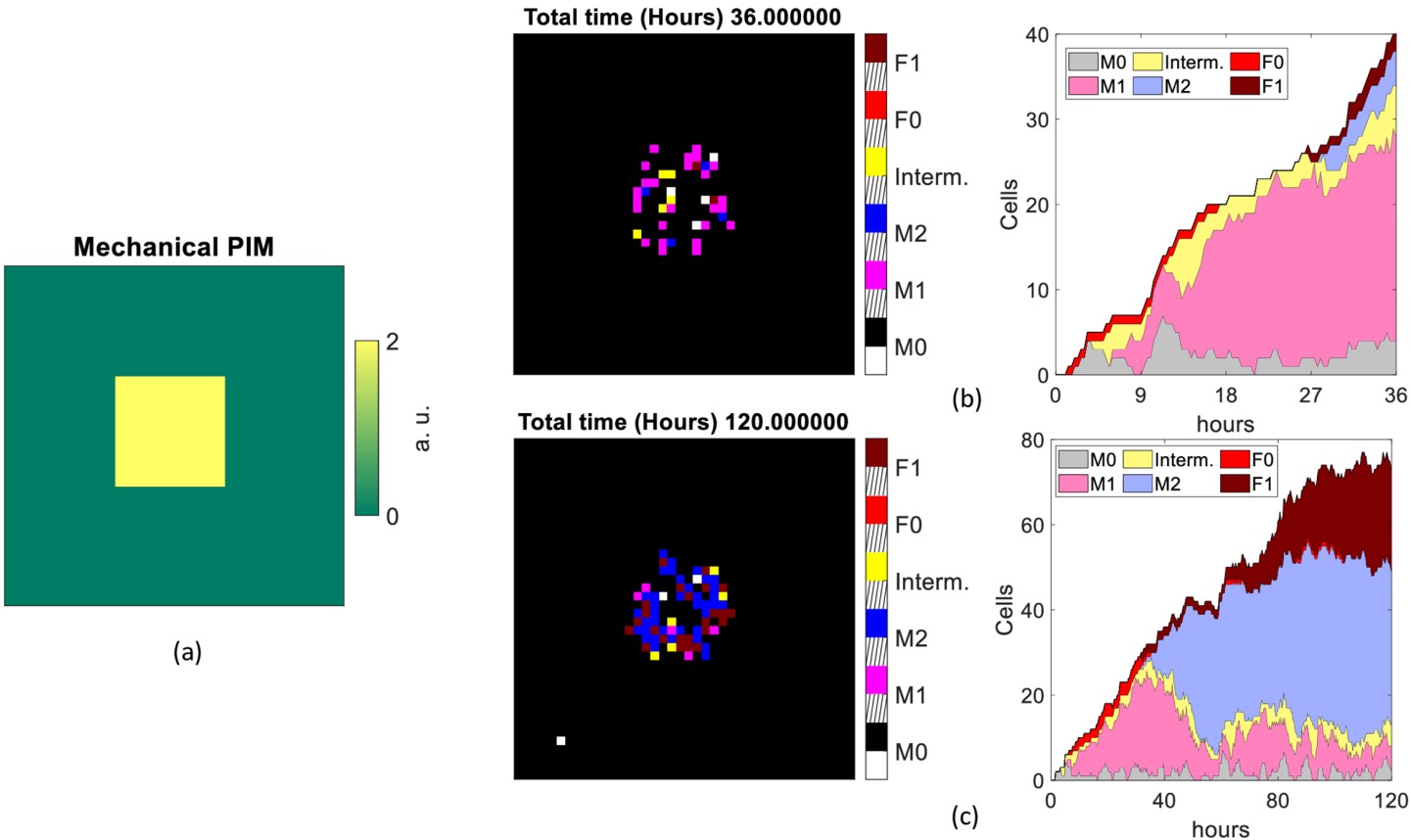

**Fig 9. Scenario 2A. Cell population distributions assuming no resident macrophages and a mechanical stimulus as input.** Initial distribution of mechanical PIM used (a). The single last frame (left side) and the overall number of cells counted at each time of the simulation (right side) lasting 36 hours (b) and 120 hours (c).

the chemical stimulus, as can be judged from the analysis of the 2.B scenario in the following paragraph.

**3.4.4 Scenario 2.B: Mechanical PIM and 400 resident macrophages.** The last scenario considered assumes 400 tissue-resident $M_0$ macrophages in input under a constant value PIM encoded in the central portion ($13 \times 13$ pixels) of the volume and representing a biomaterial. At $t = 36$ h (Fig 10 (b), left side), a provisional capsule, essentially made of $M_2$ macrophages, is found over the central biomaterial. However, the tissue surrounding the biomaterial is mainly occupied by intermediate and $M_0$ macrophages. On the other side, after 120 hours of incubation, the pixels surrounding the biomaterial are essentially empty and the central capsule is formed by $M_2$ and $F_1$, as observed in the scenario 2.A (Fig 9 (c), left side). However, the cell kinetics is remarkably different from the chemical stimulus cases (Figs 7 and 8). The cell population increases very rapidly, mainly in terms of $M_0$ cells that are directly recruited and then in terms of $M_1$ at first, and $M_2$ at a later stage (appearing after 10 hours approximately, see Fig 10 (b), right side). From 60 hours on, the major components of cells are $F_1$ and $M_2$, as observed in the cases 1.A and 1.B. This behavior substantiates our suggestion that the response to a biomaterial is delayed with respect to the prompt chemical stimulus and induces large fractions of $F_1$ and $M_2$ macrophages in the chronic stage.

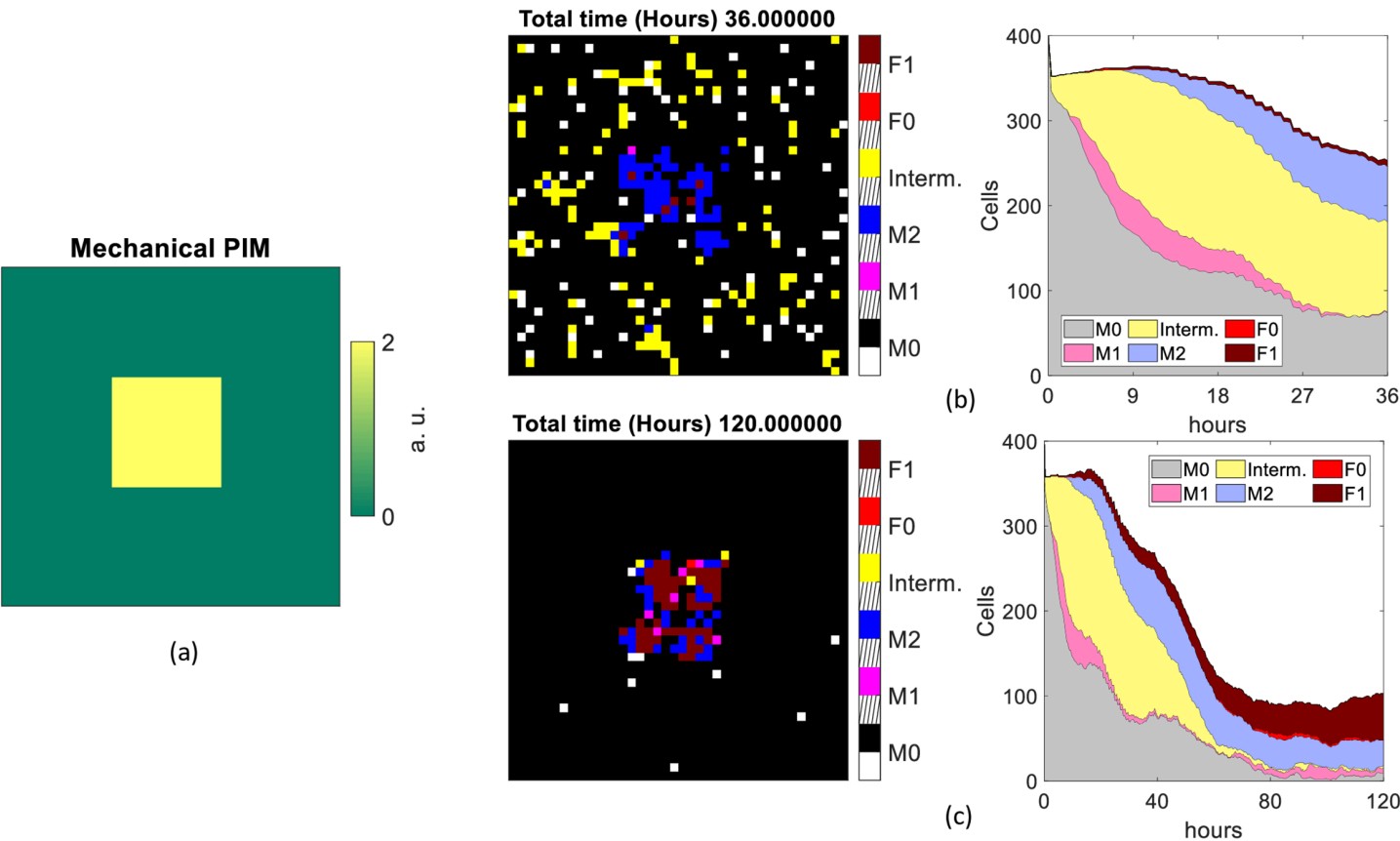

**Fig 10. Cell population distributions assuming 400 resident macrophages and a mechanical stimulus as input.** Initial distribution of mechanical PIM used (a). The single last frame (left side) and the overall number of cells counted at each time of the simulation (right side) lasting 36 hours (b) and 120 hours (c).

## 3.5 Effect of chemotaxis on the spatial cell distribution

In order to evaluate the impact of chemotaxis over distribution of cells in the grid, we compared five scenarios which differ from each other by the value of the chemotactic parameter $k$ introduced in Sect 2.2. The five values considered were $k = 0$ (no chemotaxis), $k = 0.02$ and $k = 0.5$ (the two values experimentally derived from literature), $k = 1$ and $k = 2$. To do so, we decided to run the MFF model assuming 400 tissue-resident macrophages under pathological stimulus, since in our proposal the equation of chemotaxis does not depend on the mechanical PIM. Fig 11 shows the last frames after $t = 36$ h (top row) and $t = 120$ h (bottom row), resulting from simulations launched with the selected values of the chemotaxis parameter $k$. We observe that, as the $k$ value increases, most of cells (especially $M_2$, $F_1$ and $F_0$) are concentrated in the center of the grid, where the chemical PIM is initially located. This is due to the larger attraction exerted by the stimulus in the center of the grid and it is found both for 36 h and 120 h simulation times.

Fig 12 reports the number of cells $N_c$ that change their location (pixel) over the grid under the effect of chemotaxis, as a function of $k$. It should also be noted that each cell may move more than once, since we get values of $N_c$ higher than the maximum number of cells that can be placed in the grid, coinciding with the number of pixels ($N = 1600$). For both simulation times, $N_c$ increases as the chemotactic parameter $k$ rises, as expected. The data are well described by a power law $N_c \approx A \cdot k^m$. The steepness is similar for the two simulation

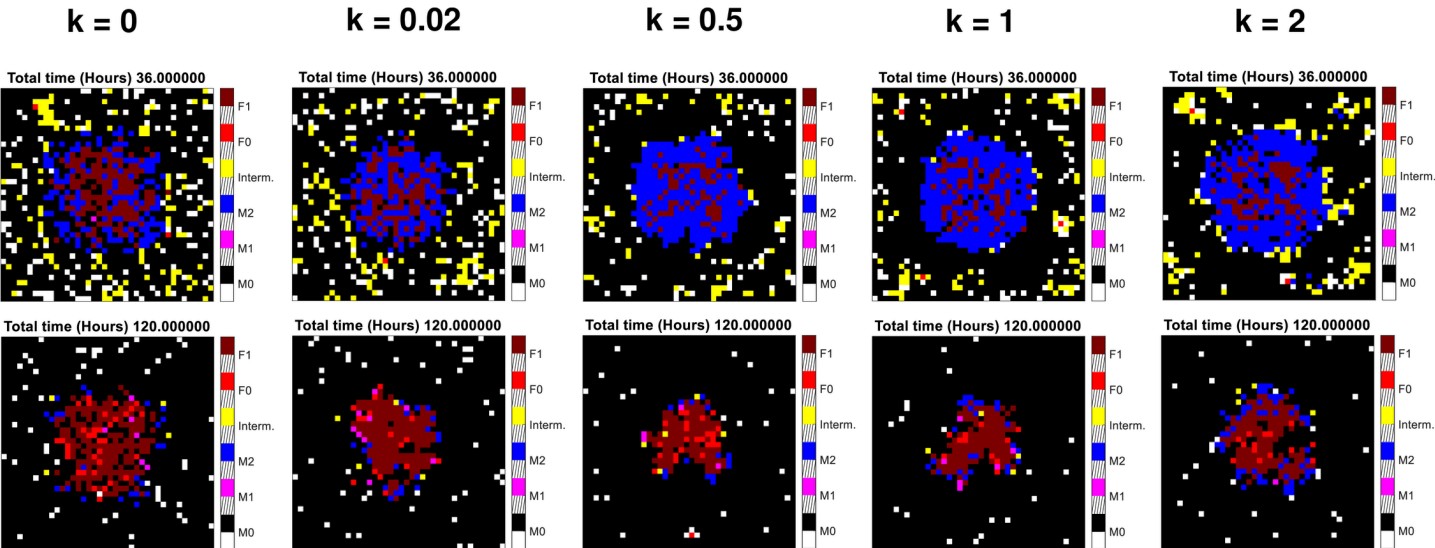

**Fig 11. Evaluation of chemotaxis over cell distributions.** From left to right: last frames of one single simulation launched using increasing chemotaxis rates from $k = 0$ to $k = 2$, for 36 (top row) and 120 (bottom row) hours. Legend of cell populations is the same used from Figs 7 to 10.

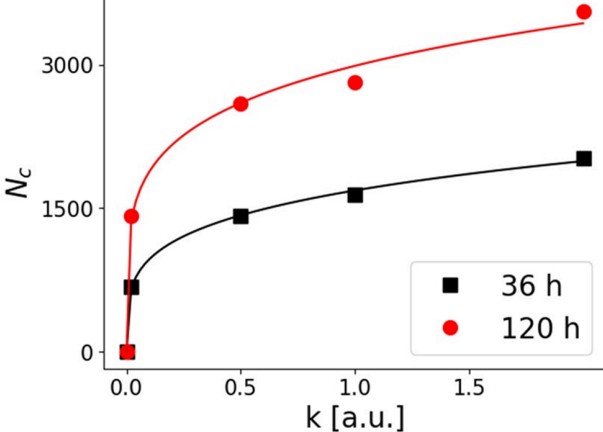

**Fig 12. Number of cells $N_c$ moving for chemotaxis** after 36 h (black squares) and 120 h (red dots) of simulation as a function of the $k$ value. The solid lines are best fit to the data with the function $N_c \approx A \cdot k^m$. Best fit values are $A = 1680 \pm 20$, $m = 0.24 \pm 0.01$ and $A = 2990 \pm 70$, $m = 0.20 \pm 0.02$.

durations, $m = 0.22 \pm 0.02$. However, the absolute value of the saturation $N_c$ value is larger after 120 hours than after 36 hours of run.

## 4 Discussion

Macrophages, fibrocytes and fibroblasts are some of the key cells in the response to many inflammation conditions triggered by a pathogen [14–16], as well as in a FBR occurring when a biomaterial is implanted [17–19]. In literature, different modeling approaches have been proposed to reproduce the immune response to an inflammatory stimulus, including differential equation based models [3,7,8], and AB schemes [3,20,21], as well as a

combination of both approaches [22]. Nevertheless, as far as we know, none of them involves different cell populations and, simultaneously, the presence of an implanted biomaterial triggering the FBR. In this work, we improved the AB model proposed in [3], the $Minucci_{ab}$ model, where macrophages were stimulated with a chemical PIM simulating LPS, by adding fibrocytes, fibroblasts, and by introducing the presence of a mechanical stimulus to reproduce the implant. Our long-term aim is to develop a digital twin of FBR to be coupled with histology, using as much available biological data as possible.

Validation performed with *in vitro* experimental data revealed a better concordance of the AB models ($Minucci_{ab}$) compared to the ODE approaches ($Maiti_{ode}$ and $Minucci_{ode}$). This result, along with recognizing that other cell types interact with macrophages through various cytokines and are essential for recapitulating FBR, led us to enrich the original AB model in [3] with these fundamental yet previously missing components.

An additional change with respect to the literature [3,8], deals with the source of the stimulus triggering the response. Our goal being the FBR by biomaterials, we introduced a "mechanical stimulus", reproducing an implanted biomaterial. It has been located in the same position in which the chemical stimulus was initially placed inside the grid, and triggers the release of pro- and anti-inflammatory cytokines. Within this framework, fibrocytes and fibroblasts are modeled as mobile agents, like macrophages, and recruited depending on the level of existing mediators, finally forming fibrotic tissue as a chronic response to inflammation. Additionally, we also wanted to include the chemotactic migration of the cells involved, which is predominant with respect to their diffusion in *in vivo* conditions. To do so, we assumed a linear dependence of the chemotaxis probability with the PIM concentration gradient, and estimated the proportionality parameter *k* with experimental data and dimensionless forward migration index found in literature. From the evaluation of four different *k* values performed in Sect 3.5, we observed that the number of cells moving by chemotaxis grows with the chemotactic parameter *k* and simulation time. Nevertheless, it should be noted that *k* has been estimated using real data from *in vitro* experiments [10], whereas in this study we are simulating *in vivo* conditions. Finally, it must be noted that chemotaxis, as it is implemented here, can be also used in a more extended model in which also the first line reaction of granulocytes to inflammation is taken into account.

We also compared results from our simulations with experimental datasets available in literature, finding good agreement in terms of cytokine production (Fig 5). In particular, we found statistically significant differences between average PIM (AIM) concentrations secreted in control and under pro-inflammatory (anti-inflammatory) stimulus, like observed in [13]. Nevertheless, we also found significant differences between average PIM (AIM) concentrations produced in control and under anti-inflammatory (pro-inflammatory) stimulus, unlike what was observed in [13]. We found that PIM and AIM kinetics reach a dynamic equilibrium at the end of the simulation time, which can indicate the immune system's ability to balance inflammatory responses in preventing tissue damage while still maintaining some activation to defend itself against infections.

Regarding the MFF model under mechanical stimulus, we found trends similar to the experimental data reported in [7] for the number of macrophages and fibroblasts counted in the areas surrounding the implant after 7 days. The number of macrophages decreases as the observation region gets closer to the biomaterial, even though almost no cells are found in the immediate vicinity of the biomaterial for our simulation conditions. There was also overall consistency in terms of fibroblasts' increase over longer times (28 days). From the experimental point of view we notice that the finding reported in [7] that the density of fibroblasts is almost constant approaching the biomaterial site, is not completely consistent with the fibrotic capsule expected in chronic conditions. On the other hand, even though we account in the

model for cell recruitment (from blood and lymphatic vessels) as triggered by the secreted PIM, we still observe lower fibroblasts' concentration close to the biomaterial at 28 days of incubation than experimentally reported [7]. This discrepancy is attributable to any of the intrinsic features of the model, including among others, the average lifespan of fibroblasts, the lower number of cells considered, the less space available and the absence of excluded volume constraints between cells and the biomaterial. Since our model does not implement an excluded volume constraint between the biomaterial and the immune cells, these tend, for long-term implants, to cover the biomaterial leaving free its direct vicinity. Despite of these model-related factors, our simulations still show that fibroblasts reach the biomaterial and persist on its surface after 28 days. The number of macrophages decreases in time, as expected. As a final remark, it must be noted that in [7], specific pancreatic macrophages were counted. Therefore, they represent a subcategory of macrophages which should be considered as a reference carefully.

An important result emerging from the evaluation of the MFF model in the four different scenarios considered in Sect 3.4 is that the response triggered by a chemical stimulus is faster (reaching a peak after twenty hours approximately) than that observed in the presence of a biomaterial, and involves all the cell populations from the first time frame. After the cell population has reached a maximum expression at about 120-150 hours, a slow decay is observed. At this stage and in the long term (120 hours), fibroblasts are the dominant cells, indicating the onset of chronicity. This occurs with and without the tissue resident macrophages, although fibroblasts are more widely expressed than M2 macrophages when simulations start with a resident $M_0$ population.

On the other hand, the mechanical PIM triggers a later response by newly recruited cells (scenario 2.A in Sect 3.4), which are concentrated on the surface of the biomaterial (Fig 9(b) and (c), left side), both over short and long simulation times, thus well recapitulating a chronic effect. The role of the recruitment of new $M_0$ and $F_0$ cells is exemplified by the observation that when resident $M_0$ cells are present, the slow increase (Fig 9) of cell density is not observed and we detect a prompt reaction (Fig 10). In addition, when resident $M_0$ cells were present, we observe the formation of a hollow polar crown around the biomaterial, which becomes more evident reducing the dimension of the implant and at intermediate frames between $t = 36$ h and $t = 120$ h (as we explored in Sect 2.3.1).

Finally, it should be noted that the MFF model proposed in this work can be adapted, with an appropriate tuning of the parameters and by adding specific tissue-related features, to reproduce various types of inflammatory and fibrotic conditions.

## 4.1 Conclusions

The MFF model proposed in this work, supported by the quantitative and qualitative evaluations conducted, is able to reproduce the dynamics and the spatial distribution of the cells primarily involved in an immune response, as well as biological processes like chemotaxis, recruitment, death of cells and formation of fibrotic tissue. These processes can be used in an extended version of this model to simulate other cell populations, such as granulocytes recruited as a first line of reaction to implants, wounds, or pathogens. In addition, an extended palette of inflammation scenarios should be considered, including hypoxia conditions, as already done in [23].

Despite these relevant achievements, the present model (MFF) is subject to a number of limitations, both technical and fundamental in nature. First, the simulations are run on a bidimensional grid. This represents an intrinsic limitation of the AB model, both in its original (*Minucci$_{ab}$*) and our improved (MFF) version. This limitation can be easily worked

around. A more fundamental limitation stems from the lack of adhesion of the fibroblasts to the biomaterial. At this stage, they were more similar to floating cells. Overcoming this limitation will be essential to better recapitulate the FBR to biomaterials that triggers cell adhesion. Our model proves to be able to reproduce immune response kinetics, but future work should include finer tuning of the model parameters (validating them with experimental data) and more robust evaluation of chemotaxis over several simulations. Finally, we did not include angiogenesis or vascularization of the biomaterials, which play a key role in the tissue remodeling process after implantation [24], and therefore should be considered in a more comprehensive FBR model. Therefore, a natural extension of the current MFF model is obviously the incorporation of those cell populations involved in new tissue formation and angiogenesis (e.g. endothelial cells).

This work is a pilot study for our long-term goal of developing a digital twin of FBR. Indeed, the MFF model, that we developed here having the FBR in mind, could also be used as a starting point to develop a digital twin of other pathological conditions. In fact, since the core of our AB model works on the interactions among different cells and populations of cells, it has the potential to be further extended to describe other specific diseases. Among them, atrial fibrillation, for the relationship of that arrhythmia with fibrotic tissue in the atrium; autoimmune diseases, which are characterized by infiltration of immune cells and, in some cases, by massive proliferation of fibroblasts [25,26]; and cancers, for the interplay between immune cells and fibroblasts in the tumor microenvironment, which can either inhibit or promote its progression.

In conclusion, the macrophages-fibroblasts model that we have developed here is a promising tool for future developments in personalized medicine and in biomaterial testing.

## Supporting information

**S1 File. Sensitivity analysis and influence of different experimental setups.**
(PDF)

## Author contributions

**Conceptualization:** Luca Presotto, Giuseppe Chirico.

**Funding acquisition:** Giuseppe Chirico.

**Investigation:** Jennifer Riccio, Luca Presotto, Shir Bahiri, Liad Doniza, Donato Inverso, Laura Sironi, Uri Nevo, Giuseppe Chirico.

**Methodology:** Jennifer Riccio.

**Software:** Jennifer Riccio, Luca Presotto.

**Supervision:** Luca Presotto, Giuseppe Chirico.

**Writing – original draft:** Jennifer Riccio.

**Writing – review & editing:** Jennifer Riccio, Luca Presotto, Shir Bahiri, Liad Doniza, Donato Inverso, Laura Sironi, Uri Nevo, Giuseppe Chirico.

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
