## [Decision Letter · Decision Letter 0]

13 Jun 2025

PONE-D-25-25184Agent-Based Modeling of Macrophage-Fibroblast Interactions in the Immune Response to BiomaterialsPLOS ONE

Dear Dr. Presotto,

Thank you for submitting your manuscript to PLOS ONE. After careful consideration, we feel that it has merit but does not fully meet PLOS ONE’s publication criteria as it currently stands. Therefore, we invite you to submit a revised version of the manuscript that addresses the points raised during the review process.

**ACADEMIC EDITOR'S COMMENTS:** 1. "Foreign body reaction (FBR)": full term for this abbreviation should be given at the first appearance in the main text. This applies to other abbreviations.

2. Lines 541-543: "Macrophages, fibrocytes and fibroblasts are some of the key cells in the response to many inflammation conditions triggered by a pathogen, as well as in a FBR occurring 

when a biomaterial is implanted": there are no references to support this statement. More references should be cited, with this one (PMID: 38793665) as an example (citing is optional).

We look forward to receiving your revised manuscript.

Kind regards,

Benjamin M. Liu, MBBS, PhD, D(ABMM), MB(ASCP)

Academic Editor

PLOS ONE

Journal Requirements:

Reviewers' comments:

Reviewer's Responses to Questions

**Comments to the Author**

1. Is the manuscript technically sound, and do the data support the conclusions?

Reviewer #1: Yes

2. Has the statistical analysis been performed appropriately and rigorously? 

Reviewer #1: Yes

3. Have the authors made all data underlying the findings in their manuscript fully available?

Reviewer #1: Yes

4. Is the manuscript presented in an intelligible fashion and written in standard English?

Reviewer #1: Yes

5. Review Comments to the Author

Reviewer #1: 

In Maiti ode: “Three sets of data obtained for 0, 0.1 and 1 µg/mL LPS stimulations are used for parameter estimation, and the fourth dataset for 10 µg/mL LPS stimulation is used for model validation”. Could the authors clarify the rationale for selecting 1 µg/mL LPS as the reference condition?

Figure 3: The FACS data appear to include only six values. Are these values from technical or biological replicates? Clarification is needed.

It would also be valuable to simulate conditions where LPS is washed out. As the current model only simulates conditions with continuous LPS exposure, it does not provide insights into the immune response following LPS removal. Simulations extending beyond 12 hours, especially post-stimulus, would be appreciated.

Biological Variability, when validating, are cells all murine cells? cell heterogeneity would benefits the model by modeling an “average” cell or system . Additionally, how does the model distinguish biological noise (natural variability in cytokine production or cellular response) from technical noise (e.g., experimental error, measurement artifacts)? A discussion on how this was handled—e.g., through replicates, probabilistic rules in the agent-based model, or sensitivity analysis—would strengthen the validation and interpretation of model robustness.

The current validation relies heavily on results from a limited source. To improve robustness, the model should be tested against a broader range of experimental datasets. This will help ensure the model is not overfitted to a specific scenario and performs reliably across diverse conditions.

Overall, this model is a good step but need more improvement on feeding more experimental data and avoid overfitting to one situation. The future of this model is bright but current immune profile of FBR can be done by wet lab alone.

6. PLOS authors have the option to publish the peer review history of their article (what does this mean?). If published, this will include your full peer review and any attached files.

Reviewer #1: No

---

## [Editor Report · Decision Letter 1]

14 Jul 2025

Agent-Based Modeling of Macrophage-Fibroblast Interactions in the Immune Response to Biomaterials

PONE-D-25-25184R1

Dear Dr. Presotto,

We’re pleased to inform you that your manuscript has been judged scientifically suitable for publication and will be formally accepted for publication once it meets all outstanding technical requirements.

Kind regards,

Benjamin M. Liu, MBBS, PhD, D(ABMM), MB(ASCP)

Academic Editor

PLOS ONE
---

## [Editor Report · Acceptance letter]

PONE-D-25-25184R1

PLOS ONE

Dear Dr. Presotto,

I'm pleased to inform you that your manuscript has been deemed suitable for publication in PLOS ONE. Congratulations! Your manuscript is now being handed over to our production team.

Kind regards,

on behalf of

Dr. Benjamin M. Liu

Academic Editor

PLOS ONE